# Privacy and security concerns with passively collected location data for digital contact tracing among U.S. college students

**Sara Belligoni**[1], **Kelly A. Stevens**[2]*, **Samiul Hasan**[3], **Haofei Yu**[3]

1 Department of Human Ecology, Rutgers University, New Brunswick, New Jersey, United States of America,
2 School of Public Administration, University of Central Florida, Orlando, Florida, United States of America,
3 Department of Civil, Environmental and Construction Engineering, University of Central Florida, Orlando, Florida, United States of America

* Kelly.Stevens@ucf.edu

**Data Availability Statement:** Data cannot be shared according to our approved IRB application. To request the data, contact irb@ucf.edu.

## Abstract

People continue to use technology in new ways, and how governments harness digital information should consider privacy and security concerns. During COVID19, numerous countries deployed digital contact tracing that collect location data from user's smartphones. However, these apps had low adoption rates and faced opposition. We launched an interdisciplinary study to evaluate smartphone location data concerns among college students in the US. Using interviews and a large survey, we find that college students have higher concerns regarding privacy, and place greater trust in local government with their location data. We discuss policy recommendations for implementing improved contact tracing efforts.

## Introduction

During the COVID-19 pandemic, smartphones became an essential tool to implement digital contact tracing procedures from local and federal governments. However, smartphone users became progressively more concerned about digital privacy and security when using apps requiring location sharing, such as those used for contact tracing, despite the potential benefits to improving health outcomes for society [1]. In order to improve government efforts for digital contact tracing using smartphone location data, we investigate the privacy and security concerns for college students with location tracking apps using a mixed methods approach [2, 3]. We use both semi-structured focus groups and a survey of 1,000 college students across the United States to gauge perceptions around security and privacy with digital contact tracing programs and use both qualitative and quantitative methods to evaluate the results.

This research builds on previous studies that evaluate characteristics, download percentages, and performances of digital contact tracing primarily administered by government programs in different countries and populations [4]. These studies find that privacy concerns have driven the public towards a lesser use of these apps, even in the view of facing a public health emergency such as that caused by SARS-CoV-2 [1]. This study focuses on the college student population for several reasons. First, most recent studies investigating location sharing data,

**Funding:** This research was supported by the University of Central Florida (UCF)'s Interdisciplinary Research (IR) Award titled as "Tracking Community Transmission and Exposure of an Infectious Disease using Passively Collected Location Data." Kelly A. Stevens (KS), Samiul Hasan (SH), and Haofei Yu (HY) are the authors who received such funding. Co-author Samiul Hasan was supported by the National Science Foundation grants CMMI 1917019. The funders had no role in study design, data collection and analysis, decision to publish, or preparation of the manuscript.

**Competing interests:** The authors have declared that no competing interests exist.

specifically for contact tracing purposes, focus on the general population in a country or sub-national jurisdiction. College students belong to a generation largely exposed to technology. Their knowledge and wide use of technology can highlight whether there is or is not mistrust towards location sharing due to privacy or security concerns. Additionally, college students have been severely affected by the COVID-19 pandemic as their educational experience has been disrupted by the transition to remote learning and the lack of campus experience. Contact tracing is being widely used as a measure to re-open campuses 'safely' throughout the country while offering in-person classes [5, 6].

However, there are many, sensitive concerns about sharing digitally-collected data with the government, especially considering recent events. In the United States, the overturn of Roe v. Wade by the U.S. Supreme Court in 2022 led to heightened concerns from privacy experts about the use of period-tracking apps that could potentially be used against a user in a judicial proceeding [7, 8]. Citizen data has also been increasingly collected and used to assist with training and development of artificial intelligence (AI) for government operations, raising numerous privacy concerns and discussions on government best practices for using digitally-collected citizen data [9].

In this study we investigate the perception of college students towards sharing location data for contact tracing purposes to see whether privacy or security concerns are more important than the goal of eventually using these resources to track the spread of COVID-19 and mitigate its effects on their college experience. Because of the complex nature of privacy and security concerns of passively collected location data, we adopted a multi-method approach. We start by collecting study participant's smartphone and GPS device locations over a one-week period from participants based in the United States. After collecting this data, we conducted semi-structured interviews with a sample of participants to better understand their privacy and security concerns regarding the collection of their location data. We ask questions that explore the relationship between trust in government and willingness to share location information.

Using this feedback, we developed a survey that was completed by over 1,000 college students across the US for their perceptions and concerns with digital contact tracing. We use qualitative and quantitative measures to evaluate the multiple choice, Likert, and open-ended questions on perceptions and use of digital contact tracing programs. We find that students have higher concerns regarding digital privacy compared to security, and place higher trust in local governments with their location data when compared to state, federal, non-partisan, or private institutions. Of all the variables evaluating demographics, location, college experience, and perceptions of security and privacy, only the institution type, work status, and health care status were statistically significant indicators of the likelihood of sharing location data for digital contact tracing purposes. We take this information and develop policy recommendations for future government-run contact tracing programs or programs that use digitally-collected location data and discuss the technical and financial challenges to implement them.

## Literature review

Contact tracing is being used in several countries, including the United States, as a measure to mitigate the spread of the COVID-19 virus. Primarily used during the first phases of countries' *reopening*, contact tracing has received mixed feelings about its effectiveness from both governments and the population. By making use of app-based technologies, governments have implemented contact tracing to track infections, possible exposures, and consequently spread during the COVID-19 pandemic. However, in order to do so, these app-based technologies heavily rely on passively collected location data that were ultimately shared, via the app, with government agencies managing the pandemic response [10]. Applications for digital contact

tracing can improve the pandemic response, and effectively prevent or identify pandemic outbreaks [1, 10].

Scholars have conducted comparative analyses across countries about the likelihood of the population downloading and using digital contact tracing. In most cases, the likelihood of people using digital contact tracing was related to their trust in the government. Countries with higher trust in the government emerged as more concerned about public health and more likely to use digital contact tracing [11]. For example, in South Korea, there have been a series of events where information about individuals testing positive for COVID-19 were made publicly available, compromising their privacy while also increasing cases of hate speech and crime against minorities [12]. This suggests that human interactions with technology can be influenced by the institutional and political context for their use, an aspect that needs further attention from the literature. An ethical design of technologies that collect citizens' data is seen as one way to mitigate privacy concerns. When governments are actively promoting measures to protect users' data, users are more likely to share digital information [9–13].

Some literature has found that concerns about the spread of the virus coupled with trust in the government influence adoption of digital contact tracing. In a multi-country comparative survey analysis conducted in Australia, France, and the United States, researchers found that where concerns about the spread of SARS-CoV-2 are higher, people were more concerned about their privacy and consequently reduced their likelihood to download and use digital contact tracing [14]. Thus, the authors of a study conducted in the United Kingdom suggest that the likelihood of the population using digital contact tracing strongly depends on their knowledge of both the pandemic and how the digital contact tracing function [15]. Similarly, Kummitha [16] analyzed potential outcomes in using technologies versus a more human approach to track the pandemic, such as through individual phone calls to collect contact tracing data. They find the human approach for pandemic response and tracking to be more effective, but also more time and resource intensive.

There are a limited number of studies on college students and contact tracing specifically. One informative study by Karosas and Ye [5] evaluates contact tracing procedures and challenges at a large, urban US college institution. They find that despite being considered an at-risk population because of the potential contact with several people every day, college students were less likely to follow guidelines, and consequently less likely to share information because of the fear of consequences [5]. As we point out earlier, college students are generally more exposed and knowledgeable about technology compared to older generations. For the general population, digital contact tracing may not have been used much because of a lack of user friendliness, however this explanation may not apply to college students more familiar with smartphone technology [17]. This, among other reasons, makes the college students population very interesting to study. Our survey of college students across the United States focuses on issues of trust, concerns about the virus, and understanding of digital contact tracing. The findings in this paper add to the literature on college student perceptions towards passively collected location data sharing for contact tracing purposes.

## Method and sample

The research presented in this article is part of a larger, interdisciplinary study evaluating the accuracy of Google Maps Location History (GMLH) and iPhone Significant Location (iSL) for contact tracing purposes. Google Maps is one of the most used navigation applications on smartphones [18–20]. When 'Location History' is enabled on the app, the smartphone user's location data will be continuously and passively collected and archived. Results from past studies have shown that GMLH and iSL data could be available for a substantial portion of the

general population for an extended time period [18–21], thus highlighting their potential for contact tracing.

In the location data accuracy study, between November 2021 and April 2022, we recruited 43 students from the University of Central Florida (UCF) to carry a GPS device with them to compare the accuracy of their smartphone location data to the reference GPS data. From this group of students, we hosted focus groups in December 2021 in order to develop a broader survey for college students. We had two groups of 6–7 students participate in a one-hour long, virtual focus group to explore their contact and location tracking data familiarity and concerns. The focus group conversations were transcribed and coded based on inductive and deductive coding. The focus groups revealed that most (92%) of the students were aware of digital location tracking before the study, and approximately a third of them (38%) turned off digital location tracking after the study concluded. The reasons for why they either left location tracking on or turned it off varied, with concerns shared regarding privacy and security of their online data. The focus group feedback was primarily used to develop an online survey for a much larger sample of students that were not involved in the smartphone and GPS device tracking portion of the study.

The online survey developed using the feedback from the focus groups was administered during the months of July and August 2022 to 1,016 full and part-time college students in the United States through Qualtrics. The survey contains 21 questions collecting demographic data, perceptions about online security and privacy, and trust in government regarding collection and use of online location data for contact tracing purposes. All survey participants are over the age of 18 and passed several speed and quality control checks conducted by Qualtrics. Additionally, Qualtrics imposed several demographic quotas to make sure the sample was representative of the overall student population in the United States. The demographic data, college, housing, employment, and health insurance status responses are summarized in Table 1.

As a quality and speed check on survey responses, we include the following definitions of privacy and security in the survey (Fig 1) and required respondents to correctly identify that using strong passwords is an example of digital security. Those who did not answer the question correctly did not have their responses recorded and were not included in our survey results. A total of 1,016 respondents answered this question correctly and their results were recorded for the survey.

We conducted content analysis using an inductive and deductive coding on open-ended question responses from the survey. The content analysis was performed manually and via the software NVivo. We analyzed the open-ended answers based on the response to the following question, "*Would you be willing to share your Google Maps Location History or iPhone Significant Location History, or other smartphone location data, for contact tracing purposes*?" Respondents were forced to answer this question in order to remain in the survey and the two response options were either "*Yes*" or "*No.*" Last, we analyze the open questions by looking at the responses from all survey participants compared to two subgroups: medical degree students and computer science degree students. These sub-groups have been identified as those whose education and training may provide a different perspective on the issues related to contact tracing for COVID-19 and technologies.

In order to quantitatively evaluate differences in opinion and thoughts on digital contact tracing, we run a logistic regression model (logit) and probability regression model (probit) on the survey data to determine which variables were most likely to impact willingness to share location data for contact tracing, finding that results were similar across models. Since the dependent variable of interest here, willingness to share location data, is a dichotomous variable with choices "yes" or "no," either a logit or probit model is useful for determining the

**Table 1. Basic demographic information, college, work, and living situation status from survey respondents (N = 1,016).** Not all questions required a response from participants.

| Question | Options | Count | Percent |
|---|---|---|---|
| Age Range | 18–24 | 729 | 72% |
| | 25–34 | 178 | 18% |
| | 35–44 | 73 | 7% |
| | 45–54 | 17 | 2% |
| | 55–64 | 6 | 1% |
| | 65 or above | 2 | 0% |
| | Other | 8 | 1% |
| Ethnicity | Asian | 48 | 5% |
| (Choose all that apply) | Black or African American | 120 | 12% |
| | Hispanic or Latino | 92 | 9% |
| | Native Hawaiian or Other Pacific Island | 1 | 0% |
| | White or Caucasian | 663 | 65% |
| | Other | 13 | 1% |
| | Prefer not to say | 10 | 1% |
| | Multiple ethnicities | 66 | 6% |
| Gender | Female | 529 | 53% |
| | Male | 463 | 46% |
| | Other | 13 | 1% |
| College Type | Private (privately owned and run) | 243 | 25% |
| | Public (government-funded, state-run) | 748 | 75% |
| College State | Democrat | 518 | 51% |
| Political Affiliation | Republican | 458 | 45% |
| | Non-Applicable | 40 | 4% |
| Work Situation | Not currently working | 448 | 45% |
| | Work full-time summers | 70 | 7% |
| | Work full-time while attending college | 114 | 11% |
| | Work part-time during summers | 78 | 8% |
| | Work part-time while attending college | 293 | 29% |
| Health Insurance Status | No | 158 | 16% |
| | Yes | 827 | 83% |
| Housing Status | Live off campus | 667 | 67% |
| | Live on campus | 202 | 20% |
| | Live on campus during academic year only | 130 | 13% |

probability a respondent will choose to share their location data based on other factors. The regression models include variables on ethnicity, age, gender, political affiliation of the state they attend college (these variables were coded based on the 2016 Presidential Elections according to the website 270 To Win), college type, college major type, health insurance status, work status, housing situation, and level of concern regarding privacy, security, and what level of government or institution could be collecting their location data. We also run t-tests to compare the means across groups for certain questions and evaluate kernel density plots to consider the distributions of responses for further insights.

All aspects of the human-subjects research was approved by the Institutional Review Board at UCF where the research took place. The consent form for the focus group was electronically distributed and signed before the focus group meetings. The online survey consent form was provided as the first question in the online survey.

**Digital security** is the practice of protecting digital information from unauthorized access, corruption, or theft. Higher security means your digital information is protected from hackers.

**Digital privacy,** on the other hand, is control over personally identifiable information. Higher privacy measures mean less identifiable information about yourself is available to those you give permission to for viewing your digital data.

Using these definitions, which of the following practices would protect your **digital security**?

> Using strong passwords

> Limiting what information you share about yourself on social media

**Fig 1. Speed and quality check question in the Qualtrics survey.** Respondents that did not correctly answer the question with the response "Using strong passwords" were dismissed from the survey and are not included in the results.

## Results & discussion

We use the regression analysis results on the survey data to first determine which variables influence willingness to share location data for contact tracing purposes among college students. Next, we consider the differences in awareness and use of smartphone location data, concern over privacy and security, and levels of concern regarding who is collecting the data. We evaluate these differences for any patterns across age, genders, state political party affiliations, majors in college, and additional variables identified by the regression as having a statistically significant influence on willingness to share location data. We find differences across genders, college institution type, health insurance and work status, but few differences across the party affiliation in the state the student currently attending college. Most of the results discussed in this section are from the survey, as the focus groups primarily informed survey development.

## Regression results on willingness to share location data for contact tracing

The regression results to determine which variables play a key role in influencing willingness to share location data for contact tracing show only a few variables were statistically significant (Table 2). We report only the statistically significant variables at the 90% or above statistical significance level (e.g., p-value greater than 0.1). We find that respondents that attend a public institution, have health insurance, or work full or part-time were more likely to share their location data for contact tracing compared to their peers. These results were consistent across the logit and probit models. The variables with no statistical significance include ethnicity, age, gender, political affiliation of the state they attend college, college type, college major type, health insurance status, work status, and housing situation. Additionally, concerns about privacy and security or which type of institution could hypothetically collect their location data did not stand out as statistically significant indicator of willingness to share location data for contact tracing, however we evaluate these responses further in the next section.

**Table 2. Logit and probit model results for the variables with at least 90% statistical significance on willingness to provide smartphone location data for digital contact tracing purposes.**

|  | Logit | SE | Probit | SE |
|---|---|---|---|---|
| College Type—Public | 1.545* | (0.270) | 1.311* | (0.142) |
| Health Insurance—Yes | 1.511+ | (0.324) | 1.294+ | (0.172) |
| Work Full Time—Summers | 1.752+ | (0.525) | 1.413+ | (0.259) |
| Work Full Time—All Year | 1.775* | (0.457) | 1.423* | (0.224) |
| Work Part Time—Summer | 1.791* | (0.531) | 1.437* | (0.261) |
| N | 845 |  | 845 |  |

Exponentiated coefficients; Standard errors in parentheses

+ $p < 0.1$

* $p < 0.05$

** $p < 0.01$

*** $p < 0.001$

Our regression results show that having health insurance coverage increases the probability that a respondent would be willing to share location data for contact tracing purposes. Previous research has found that individuals are more likely to get tested or see a doctor—as per guidelines in place in the first phases of pandemic management—knowing that the cost would be covered by their health insurance provider [21]. Those without health insurance coverage may fear incurring high medical costs associated with testing and attending doctor's visits. Interestingly, work status is another statistically significant variable that increases the probability a respondent would be willing to share location data, indicating that individuals with a stable job are more likely to share data for contact tracing purposes. This might be the case because students with a stable job are more likely to receive a series of benefits, including health insurance and paid medical leave, which makes them able to 'afford' to get tested, see a doctor, quarantine or isolate because of the protection measures offered by the employer [21]. However, health insurance status was not correlated with work status in our study, therefore these two variables are independent (according to HealthCare.gov, in the United States, persons under the age of 26 can typically stay on their parent's health insurance plan).

Last, our regression results show that students at a public college or university are more likely to share location data when compared to students at a private institution. Public and private college institutions can differ in terms of tuition costs, state versus out-of-state enrollment, and financial incentives, which may impact likelihood of reopening for in-person classes during the COVID-19 pandemic [6–20]. Previous research finds that there is some evidence that public universities, compared to private ones, were more likely to resume classes online rather than in-person during the year of 2020 after the COVID-19 pandemic started [21]. While the majority of our respondents (75%) report attending a public institution, it is possible that the differences in reopening decisions influenced student responses in our survey regarding willingness to share location data for contact tracing. Perhaps students at public universities, that were online right after the COVID-19 pandemic, were more familiar with contact tracing programs.

Previous research also finds that universities located in states with a high Republican leadership in state government were more likely to reopen in person. In our regression, there is no statistically significant difference based on party affiliation of the state where the student attends college for willingness to share location data for contact tracing. To evaluate this further, we ran a t-test to compare students in republican versus democratic states. We anticipated that states with a higher democratic leadership may be more familiar and willing to

share contact tracing data compared to republican led states [13, 24]. For the most part, our results were not statistically different across states. This may be because there was limited use of digital contact tracing in the United States at the state or local level, and very few programs operated with high public participation. Further, since our survey was conducted in the summer of 2022, it is possible students had moved or were living in a different state compared to when most of the digital contact tracing programs took place at the state or local level, which was primarily in 2020 and 2021 [13, 24]. However, based on previous literature and strong political stances towards contact tracing, this topic is worth further investigation.

## Awareness and use of location data

In addition to evaluating characteristics that might affect individual's willingness to share location data for contact tracing purposes, we also asked respondents a few questions in the survey to evaluate if they are aware of smartphone location tracking, and if they keep their location data on with specific questions tailored for their phone type (e.g., iPhone vs. Android) and location app usage, providing visual guidelines on how to find the required information. All respondents were required to own a smartphone in order to take the survey, and all the questions in this section were required to be answered. We find that most of the participants own an iPhone (compared to other phone types), and most iPhone users have 'Significant Location' turned on, which records their location history (see Table 3). Further, most participants use Google Maps on their phone as well, which also uses smartphone location data. This indicates that the majority of our participants are aware of and use smartphone location tracking for some reason or another. There were no statistically significant differences across gender on location app awareness, however women, compared to men, were slightly more likely to have their location data turned on (75% women have it on, compared to 69% of men). In the focus groups leading up to the survey, several women explained that they share their digital location data with their friends for safety reasons, which might extrapolate to the larger sample of female students who completed the survey.

Next, we asked survey respondents if they would be willing to share their smartphone location data for contact tracing purposes. We find that 56% of respondents would be willing to share their location data for contact tracing, with similar proportions across men and women. This indicates that while respondents are generally aware and comfortable with recording their digital location history using smartphones, they were less likely to agree to share that data with an outside party for contact tracing.

## Privacy and security

While privacy and security concerns were not statistically significant in the regression results on willingness to share location data for contact tracing, survey respondents had a statistically significant higher concern over privacy (mean score 7.2) compared to security (6.8) at the 99.9% significance level (Fig 2). The Kernel density plots of responses for this question (Fig 3)

**Table 3. Response rates for questions on smartphone type and use of other location tracking services and apps.**

|  | Yes | No |
|---|---|---|
| Do you own a smartphone? | 100% | 0% |
| Is your phone an iPhone? | 70% | 30% |
| If yes—do you have Significant Location turned on? | 73% | 27% |
| Do you use Google Maps app on your smartphone? | 77% | 23% |

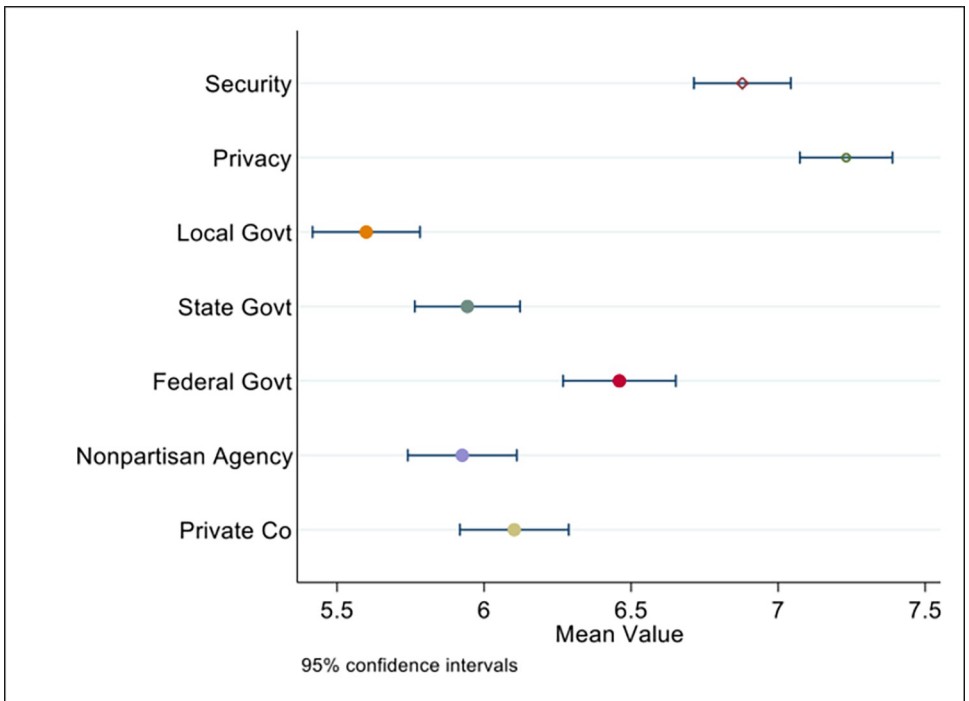

**Fig 2. Mean with 95% confidence intervals for questions rated on a scale from 0–10, with 0 meaning no concern or complete trust and 10 meaning extreme concern or complete distrust.**

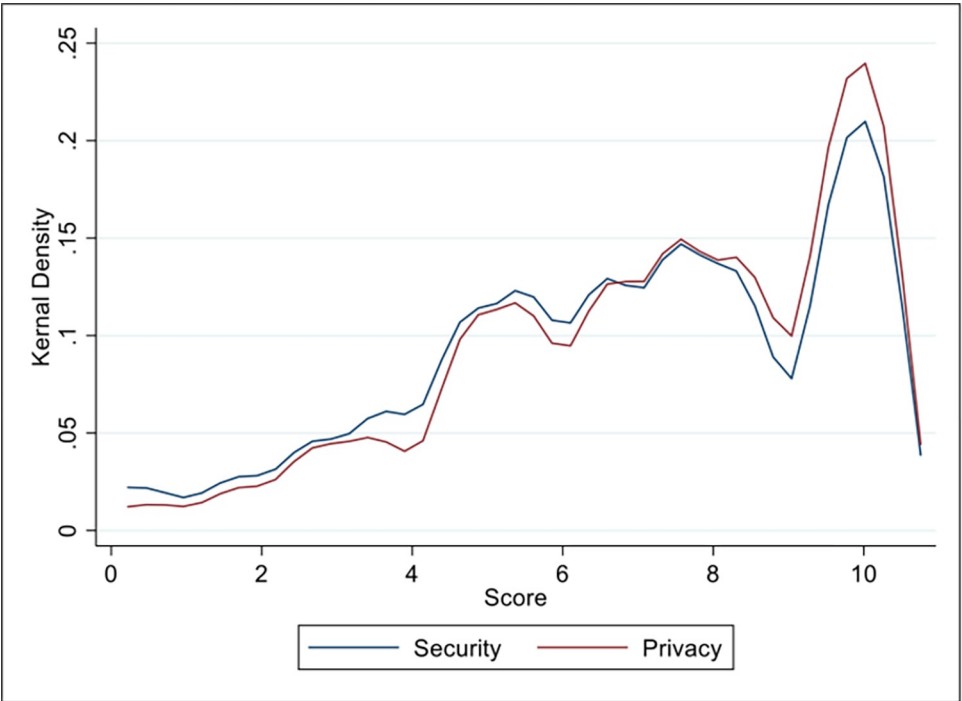

**Fig 3. Kernel density plot with bi-weight kernel function for the distribution of responses to overall concerns about privacy and security of their smartphone location data, with 0 meaning no concern or and 10 meaning extreme concern.**

reveal the response distributions are similar, however there were more respondents with high privacy concerns (score of 8–10 on a scale of 0–10, with 0 being no concern and 10 being highest concerns) compared to security. Further, women compared to men had a statistically significant higher concern over privacy, but not security, with a mean score of 7.3. Because the distributions are so similar between privacy and security, it is possible that there was little distinction between the two properties across the sample, despite everyone passing the speed check (Fig 1) that performed a simple check on their understanding of digital security.

The survey respondents who identified themselves as more likely to share location data for contact tracing report more specific reasons for concern related to privacy when compared to those who identified themselves as less likely to share location data for contact tracing. For example, from those more likely to participate in contact tracing, concerns about privacy included not wanting people to know their location, especially if it revealed information about friends and family. One respondent to the survey mentioned, "*I would not like to have the wrong people with my location information and sometimes I wouldn't want my friends or other visitors locations to be revealed especially the more private ones.*" Another respondent stressed that there can be a, "*Breach of privacy, and I don't want people knowing where my kids live.*" One of the recurring themes expressed by another interviewee is that "*Privacy is always a concern. I do not want my information to be used unethically or illegally.*"

## Trust in government

As seen in Fig 2, there were statistically significant differences across the means regarding concerns over who collected smartphone location data from respondents for contact tracing. The highest level of concern was over collection by the federal government, with a mean score of 6.4 on a scale of 0–10 with 10 indicating high concern and no trust in government with their online location data. By triangulating the statistical results with the open questions resulting from the survey conducted, respondents seem worried about the potential use of data from the government for scopes that go beyond those of contact tracing. For example, one survey respondent mentioned, "*the government may one day decide to use location information for other purposes.*" Another respondent stated, "*I am concerned that data gathered for contact tracing purposes could eventually be used to keep tabs on those who do not comply with government regulations.*"

Comparatively, the score for a private company was next highest (mean score 6.1), then state agency (mean score 5.8) as well as nonpartisan government agency (mean score 5.8). Local government had the lowest level of concern at mean score 5.5. The means were statistically significantly different for all relationships at the 95% confidence level except for state government compared to non-partisan government agencies, which were very similar mean values. From this, we can deduce that college students place highest trust in local governments collecting smartphone location data for contact tracing, and the least amount of trust in the federal government.

In the kernel density plot comparing distributions for levels of trust across government and non-government agencies (Fig 4), two main modes appear: around an average score of 5 (halfway through the scale of 0–10), and at 10 (indicating no trust). About 20% of respondents had no trust in the federal government compared to 12% of respondents having no trust in local government with their location data. Women had slightly higher mean levels of concern regarding collection of their smartphone local data across agencies, however this mean was only statistically different at the state level, with women having a mean concern of 6 compared to men at 5.6. The distribution patterns in Fig 4 are similar, however, so it is possible that survey respondents had difficulty distinguishing their level of trust across different levels of government or organizations.

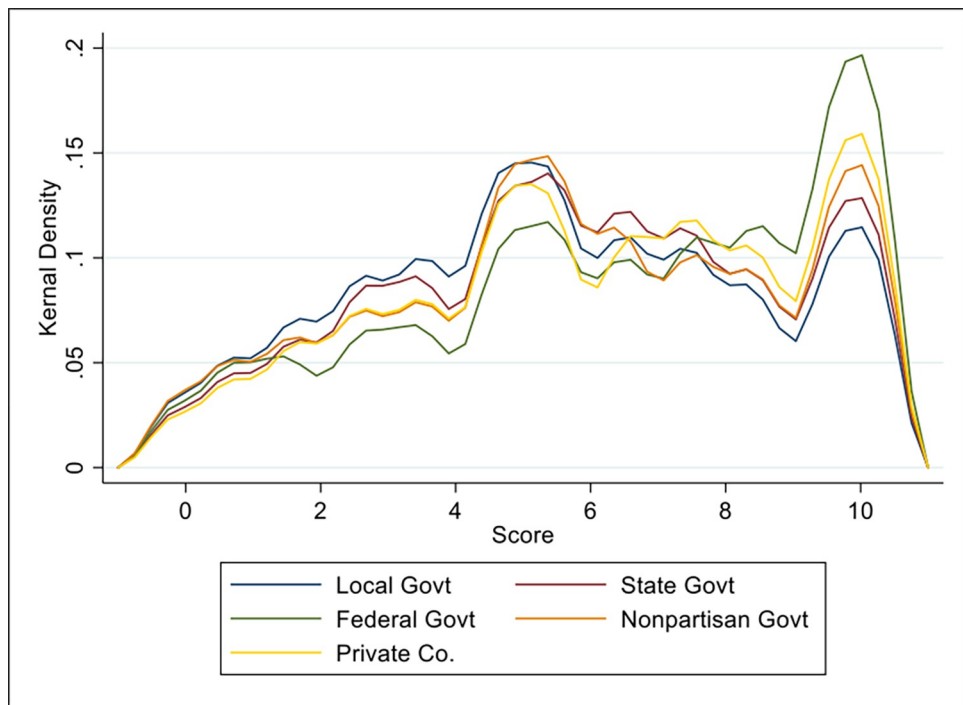

**Fig 4. Kernel density plot with bi-weight kernel function for the distribution of responses to overall trust with their smartphone location data by different organizations, with 0 meaning no concern or high trust, and 10 meaning extreme concern and no trust.**

## Field of study & political affiliation

Since college students are heavily focused in their field of study, we anticipated that sentiments regarding location data and contact tracing may vary based on their major. Further, previous research has shown that context and information are important indicators of data sharing and app usage. We broke up the sample into students in medical fields, computer and cybersecurity fields, and all other majors. About 8% of the sample were in computer or cybersecurity fields and 16% in medical fields. We did not see any statically significant differences in survey responses regarding awareness and use of location data or concerns regarding privacy and security of location data for contact tracing across different majors. However, there were some comments provided in the open-ended survey questions and the focus groups that might provide insight for future research.

An important point stressed among college students majoring in the medical field, is that, as one survey respondent said, "*Generally, location information is tricky and can certainly be a breach of privacy if used with malicious intent. That said, I do believe (if used correctly) it would be great for contact tracing for easily spread diseases.*" This suggests that students in the medical field may have concerns for their digital privacy, however, they recognize the importance and value, from a public health perspective, of using location data for contact tracing purposes.

College students majoring in computer science-related fields, instead, seem more concerned about privacy and think that contact tracing, for the way in which it is structured, may lead to unnecessary alerts. One of the survey respondents shares, "*If [contact tracing] does eventually discover that you may or may not have been in contact with someone with COVID-19, then, even if you were not infected or even close to the infected person, you would be tracked as being in contact and that information could be used by employers to keep you at home, even though you weren't even in contact.*"

On the other hand, respondents who identified themselves as less likely to participate in contact tracing seem to be very focused on their concerns about privacy and potential criminal activities associated with it as data can end up in the 'wrong hands.' For example, an survey respondents in the medical field said that *"I feel like tracing someone for a certain purpose is okay such as kidnapped or been gone for days. But to look and see where someone is at is a breach of privacy."* Another student majoring in computer science stated, *"I would not have concerns as long as my data was only used temporarily by public health officials/medical practitioners for the purpose of contact tracing."* This seems to suggest that sharing location data may be perceived relevant for specific purposes but concerns about privacy and criminals potentially accessing personal information are very important. For example, among students in computer science-related majors, some highlighted that *"Crimes could be pin pointed"* and that there is *"Potential for knowing when not home for theft."*

## Solutions

Table 4 shows frequency of different ways an organization could potentially persuade students to share their location data for contact tracing purposes based on survey responses. The most frequent response to the question, which allowed respondents to choose all that apply, was to provide a cash incentive (55%). Among other financial incentives, tax or service fee reductions scored much lower at 25%. Clear messaging and evidence of strong security measures, privacy measures, or how the data will be used all scored about 40%, along with the ability to easily opt in or out of the program, when such opportunity is given at any time. Providing information on the benefits to the individual or society scored much lower around 25%. Only 15% of respondents stated that there was no way to encourage them to share their location data, suggesting that there is room for government intervention in these spaces.

## Policy implications and conclusion

From this research, we conclude that privacy is among one of the biggest concerns regarding collection of location data for contact tracing purposes, in line with what other studies have found [2–14]. Therefore, any digital contact tracing program utilizing smartphone location data should use the latest privacy-protection measures when implementing such programs. Digital contact tracing programs should partner with smartphone companies who store the GPS location data collected from users' smartphones. Any contact tracing algorithm may run

**Table 4. Frequency of responses that included each option as one-way organizations could potentially encourage them to share their location data for contact tracing purposes.**

| Proposed Method | Frequency |
|---|---|
| Cash incentive (e.g., money or direct payment) | 55% |
| Clear messaging and evidence of strong security measures | 40% |
| Transparency and clear messaging on how data will be used | 40% |
| Ability to easily opt in or out of location tracking | 39% |
| Clear messaging and evidence of strong privacy measures | 39% |
| Guarantee that the data will not be used for any other purpose | 38% |
| Explaining the benefits to you for participation in location tracking | 25% |
| Tax or service fee reduction (e.g., pay lower taxes or lower service fees) | 25% |
| Explaining the benefits of location tracking for society | 23% |
| Clear messaging about how the location tracking data will be used | 17% |
| None- there are no ways to encourage me to share my location data for contact tracing | 15% |
| Other | 1% |

in the servers of smartphone companies to ensure user privacy. In this study, we adopted multiple approaches for protecting participants' privacy which may also be used in other digital contact tracing programs, and we were clear about these privacy measures in written consent forms and in-person explanation of research sessions. For example, all participants' identifiers were removed from the collected location data by assigning a random and unique subject ID. Only the principal investigator has access to a password-protected list where the corresponding identities of the participants are stored. All collected location data were encrypted using GNU Privacy Guard (GnuPG) tool [Ref] and stored in a server accessible to specific users. All data analysis was performed on the server and no local data copies were allowed.

Additionally, clear and transparent messaging about what data is being collected and how it can be used may enhance adoption by college students that the literature have found to be reluctant in using digital contact tracing [5]. Indeed, the main concerns with privacy focus on how the government may use data. Even among students concerned about security (more than with privacy) the reluctance in sharing data refers to the possibility of the latter to be used for criminal activities. Therefore, agencies running digital contact tracing programs will need to enhance messaging through public campaigns and app design. Agencies may need to test-run particular messages with the public to make sure they are clear and interpreted as intended. Last, contact tracing programs should be designed to allow the user to easily opt in or out of using their location data for such purposes.

Further, we find that local governments are more trusted with digital location data compared to other government institutions or private companies, which is partially in line with the scholarship that have found low levels of trust towards the government overall [9, 11, 13]. For this reason, and where feasible, local government agencies should become a partner in contact tracing programs for ensuring clear messaging on the use of location data. Thus, in the United States, contact tracing was not managed at the federal level and states were left with the opportunity to decide the model to use for contact tracing. According to the National Academy for State Health Policy in the United States, 31 States, including the District of Columbia, as of Fall 2022, have suspended any state contact tracing programs or shifted to locally run contact tracing [25]. Since local governments have more difficulty raising funding to offer financial incentives to residents, funding for incentives should come from the federal government. For example, Coronavirus Aid, Relief, and Economic Security Act (CARES Act) and other economic relief funds during the pandemic issued by Congress should have some designated use for incentives and administration of locally-run, digital contact tracing programs.

For the concerns about the government usage of data for different scopes other than contact tracing, creating legislation that would prevent government agencies from using data about location collected for contact tracing purposes for any different use than that, would help building trust. In other words, legislation that would prevent sharing these data for uses different than those for which they were collected (unless it is a matter of national security).

Since we focus on college students in this study, it is also possible that colleges administering digital contact tracing programs could be more successful with the adoption of these programs. Students *may* place a higher degree of trust in the college institution when compared to the government [22–26], however more research is needed to determine the degree to which that is true. Privacy and security concerns of a significant portion of the students should be considered before implementing such a program at a college level. Approximately 41% of colleges in the United States opted for using contact tracing technologies during the Fall 2020 semester [27], therefore more can be learned from the successes and challenges with these college-run programs. Our study provides further insights on college student concerns regarding privacy and security that may be useful for future government programs collecting digital data to improve collective health outcomes [28].

## Author Contributions

**Conceptualization:** Kelly A. Stevens, Samiul Hasan, Haofei Yu.

**Data curation:** Kelly A. Stevens, Samiul Hasan, Haofei Yu.

**Formal analysis:** Sara Belligoni, Kelly A. Stevens, Samiul Hasan, Haofei Yu.

**Funding acquisition:** Kelly A. Stevens, Samiul Hasan, Haofei Yu.

**Investigation:** Sara Belligoni, Kelly A. Stevens, Samiul Hasan, Haofei Yu.

**Methodology:** Sara Belligoni, Kelly A. Stevens, Samiul Hasan, Haofei Yu.

**Project administration:** Kelly A. Stevens, Samiul Hasan, Haofei Yu.

**Resources:** Kelly A. Stevens, Samiul Hasan, Haofei Yu.

**Supervision:** Kelly A. Stevens, Samiul Hasan, Haofei Yu.

**Validation:** Sara Belligoni, Kelly A. Stevens, Samiul Hasan, Haofei Yu.

**Visualization:** Kelly A. Stevens, Samiul Hasan, Haofei Yu.

**Writing – original draft:** Sara Belligoni, Kelly A. Stevens, Samiul Hasan, Haofei Yu.

**Writing – review & editing:** Sara Belligoni, Kelly A. Stevens, Samiul Hasan, Haofei Yu.

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
