## [Decision Letter · Decision Letter 0]

8 Oct 2023

PONE-D-23-11062Privacy and Security Concerns with Passively Collected Location Data for Digital Contact Tracing Among U.S. College StudentsPLOS ONE

Dear Dr. Stevens,

Thank you for submitting your manuscript to PLOS ONE. After careful consideration, we feel that it has merit but does not fully meet PLOS ONE’s publication criteria as it currently stands. Therefore, we invite you to submit a revised version of the manuscript that addresses the points raised during the review process.

The manuscript need only minor revisions before it can be accepted for publication.

We look forward to receiving your revised manuscript.

Kind regards,

Shadab Alam, Ph.D.

Academic Editor

PLOS ONE

Additional Editor Comments :

The manuscript need only minor revisions before it can be accepted for publication.

Reviewers' comments:

Reviewer's Responses to Questions

**Comments to the Author**

1. Is the manuscript technically sound, and do the data support the conclusions?

Reviewer #1: Yes

Reviewer #2: Yes

2. Has the statistical analysis been performed appropriately and rigorously? 

Reviewer #1: Yes

Reviewer #2: Yes

3. Have the authors made all data underlying the findings in their manuscript fully available?

Reviewer #1: No

Reviewer #2: Yes

4. Is the manuscript presented in an intelligible fashion and written in standard English?

Reviewer #1: Yes

Reviewer #2: Yes

5. Review Comments to the Author

Reviewer #1: The proposed study is an interesting an timely topic.

The main conclusion to be drawn is that there is more trust in local government than in federal government regarding the protection of their data. Still, it is more towards distrust as it is more than 5 in the scale.

- It is not clear which part of the reported results is based on the focus groups.

Additionally, the focus groups results may seem to be biased, as the students who participated on the focus group were already

giving their smartphone and GPS device locations over one week period.

Moreover, this data collection, does not seem to have any relevance or relation to the paper results.

- Having the captions of the figures in place without the figure appears to be an error, but may have been made on porpose.

- It may be relevant to know what do the surveyed students understand as security or privacy concerns,

since their answers reflected as a Kernel Density Plot (KDP) in Figure 3 security and privacy have almost the same density distribution.

This may indicate that security and privacy are commonly understood as the same/similar property.

- And again, in figure 4 the KDP seems that all variables follow the same distribution (up to scale)

The two green colors are difficult to distinguish one from the other.

Reviewer #2: The following minor revision comments need to be addressed by the author for the manuscript to be accepted.

• Source Attribution: Provide specific citations for the statistics mentioned, such as the 41% adoption rate of contact tracing technologies in colleges during the Fall 2020 semester.

• Consistent Terminology: Ensure consistent use of terminology; for example, use either "digital contact tracing" or "contact tracing apps" consistently throughout the document.

• Quantify User Trust: When stating that students might trust college institutions more, include relevant statistics or surveys to quantify this trust for added credibility.

• Explanation of Encryption Methods: Provide a brief explanation of the encryption methods employed for the collected location data for readers unfamiliar with encryption techniques.

• Specifics on Opt-In/Opt-Out Mechanism: Elaborate on the mechanisms allowing users to opt in or out. Mention if it's a one-time choice or if users can change their preferences at any time.

• Duration of the Study: Include the duration of the study or specify the time frame during which data was collected to provide context for the findings.

• Comparison with Previous Studies: Discuss briefly how the findings align or differ from previous studies on similar topics, providing a comparative perspective.

• Clarify 'Success' of College-Run Programs: Define what constitutes 'success' in college-run contact tracing programs. Is it high participation rates, effective containment of outbreaks, or another metric?

6. PLOS authors have the option to publish the peer review history of their article (what does this mean?). If published, this will include your full peer review and any attached files.

Reviewer #1: No

Reviewer #2: No

---

## [Author Response · Author response to Decision Letter 0]

20 Oct 2023

Dear Dr. Alam,

Thank you for giving us the opportunity to submit a revised draft of the manuscript “Privacy and Security Concerns with Passively Collected Location Data for Digital Contact Tracing Among U.S. College Students” for publication in Plos One. We appreciate the time and effort that you and the reviewers dedicated to providing feedback on our manuscript and are grateful for the insightful comments on and valuable improvements to our paper. We have incorporated the suggestions made by the reviewers. Those changes are highlighted within the manuscript with track changes. Please see below, highlighted in blue, for a point-by-point response to the reviewers’ comments and concerns. 

Thank you and we look forward to having our manuscript accepted for publication.

Sincerely,

Kelly A. Stevens (corresponding author)

* * *

Reviewer #1: The proposed study is an interesting and timely topic. The main conclusion to be drawn is that there is more trust in local government than in federal government regarding the protection of their data. Still, it is more towards distrust as it is more than 5 in the scale.

1. It is not clear which part of the reported results is based on the focus groups. 

We have edited the manuscript in Section “Method & Sample” and “Results” to clarify that most of the reported results are based on the survey data. The focus groups were primarily used to inform survey development. In the one place we refer specifically to an observation from the focus group (Section “Awareness and Use of Location Data”), we identify that the observation was from the focus group.

2. Additionally, the focus groups results may seem to be biased, as the students who participated on the focus group were already giving their smartphone and GPS device locations over one-week period. 

We appreciate you raising this point. For this reason, we did not rely heavily on focus group observations in the discussion of the results, and rather the focus groups informed survey development. We added in Section “Method and Sample” to clarify, “The focus group feedback was primarily used to develop an online survey for a much larger sample of students that were not involved in the smartphone and GPS device tracking portion of the study.” 

3. Moreover, this data collection, does not seem to have any relevance or relation to the paper results. 

The focus groups informed survey development, so we clarify in the paper that the results are a discussion of the survey results and not often a discussion of the focus group observations except in one case mentioned above in Section “Awareness and Use of Location Data” in Point 1 above.

4. Having the captions of the figures in place without the figure appears to be an error but may have been made on purpose. 

Thanks for your comment. As per journal guidelines, we included the figures caption in the text where they should appear. We then uploaded each figure as a separate file, properly renamed. 

5. It may be relevant to know what do the surveyed students understand as security or privacy concerns, since their answers reflected as a Kernel Density Plot (KDP) in Figure 3 security and privacy have almost the same density distribution. This may indicate that security and privacy are commonly understood as the same/similar property. 

Great point. The speed check in Figure 1 provided definitions of digital security and digital privacy and was meant to remind or educate survey takers on the differences between these two properties and remove survey takers who answered the question on digital security incorrectly. However, we acknowledge this was not a very thorough test to distinguish the two, and it is possible the similar distributions identify that these two concepts are commonly considered the same way.

We added the following to Section “Privacy and Security” where we discuss security versus privacy to address this, “Because the distributions are so similar between privacy and security, it is possible that there was little distinction between the two properties across the sample, despite everyone passing the speed check (Figure 1) that performed a simple check on their understanding of digital security.”

6. And again, in figure 4 the KDP seems that all variables follow the same distribution (up to scale) 

Similar to the comment regarding privacy and security, it is possible survey respondents have difficulty distinguishing levels of trust across different organizations. The values are often statistically different according to the t-tests of means, but we acknowledge the similar distribution by adding at the end of Section “Trust in Government,” “The distribution patterns in Figure 4 are similar, however, so it is possible that survey respondents had difficulty distinguishing their level of trust across different levels of government or organizations.”

7. The two green colors are difficult to distinguish one from the other. 

We edited the graph so private company is now gold.

Reviewer #2: The following minor revision comments need to be addressed by the author for the manuscript to be accepted.

1. Source Attribution: Provide specific citations for the statistics mentioned, such as the 41% adoption rate of contact tracing technologies in colleges during the Fall 2020 semester. 

We have included the full citation of the source, both in-text and in the references list (last paragraph of Section “Policy Implications and Conclusion”).

2. Consistent Terminology: Ensure consistent use of terminology; for example, use either "digital contact tracing" or "contact tracing apps" consistently throughout the document. 

We have edited accordingly and used the terminology “digital contact tracing” when appropriate.

3. Quantify User Trust: When stating that students might trust college institutions more, include relevant statistics or surveys to quantify this trust for added credibility. 

We appreciate your comment. In the Section “Policy Implications and Conclusion,” we have included a recent news story from 2022 including a survey addressing different perceptions of trust among students toward higher education institutions and the government. This article reports that “Overall, survey respondents trusted U.S. colleges and universities more than American media companies, corporations or the U.S. government; more than 55 percent of U.S. adults said they had "some" or "a lot" of trust in higher education.” While not a journal article, we believe this is worth further investigation and have now italicized may to emphasize that this is still in question by academic literature.

4. Explanation of Encryption Methods: Provide a brief explanation of the encryption methods employed for the collected location data for readers unfamiliar with encryption techniques. 

By encryption, we mean anonymization and password protection. We anonymize the collected location data by assigning random and unique subject IDs. All data are stored on an off-line, password-protected storage device and only the PI and the designated data analyst will have access to the data. We clarify this further in Section “Policy Implications and Conclusion” by stating, “Only the principal investigator has access to a password-protected list where the corresponding identities of the participants are stored. All collected location data were encrypted using GNU Privacy Guard (GnuPG) tool and stored in a server accessible to specific users. All data analysis was performed on the server and no local data copies were allowed.”

5. Specifics on Opt-In/Opt-Out Mechanism: Elaborate on the mechanisms allowing users to opt in or out. Mention if it's a one-time choice or if users can change their preferences at any time. 

Thank you for pointing this out. We clarified this point in the sub-paragraph “Solutions” of our manuscript to indicate the ability to easily opt-in or out at any time.

6. Duration of the Study: Include the duration of the study or specify the time frame during which data was collected to provide context for the findings. 

We have mentioned about the online survey data collection period which was July-August 2022, and we specified in the “Method and Sample” Section, the data collection period for the GPS which was November 2021-April 2022.

7. Comparison with Previous Studies: Discuss briefly how the findings align or differ from previous studies on similar topics, providing a comparative perspective. 

Thank you for this suggestion. We identified how the main themes discussed in our “Policy Implications and Conclusion” appear to be in line or different than what was discussed in the literature (mixed feelings towards digital contact tracing, trust in government, privacy concerns more than security concerns, college students not likely to adopt digital contact tracing run by academic institutions because of the fear of consequences).

8. Clarify 'Success' of College-Run Programs: Define what constitutes 'success' in college-run contact tracing programs. Is it high participation rates, effective containment of outbreaks, or another metric? 

We appreciate your comment and specified the success mainly refers to the adoption of digital contact tracing in terms of higher participation rates. We clarify this in “Policy Implications and Conclusion” section.

---

## [Decision Letter · Decision Letter 1]

2 Nov 2023

Privacy and Security Concerns with Passively Collected Location Data for Digital Contact Tracing Among U.S. College Students

PONE-D-23-11062R1

Dear Dr. Stevens,

We’re pleased to inform you that your manuscript has been judged scientifically suitable for publication and will be formally accepted for publication once it meets all outstanding technical requirements.

Kind regards,

Shadab Alam, Ph.D.

Academic Editor

PLOS ONE

Additional Editor Comments (optional):

Based on the review request, the article can be accepted for publication in its current form.

Reviewers' comments:

Reviewer's Responses to Questions

**Comments to the Author**

1. If the authors have adequately addressed your comments raised in a previous round of review and you feel that this manuscript is now acceptable for publication, you may indicate that here to bypass the “Comments to the Author” section, enter your conflict of interest statement in the “Confidential to Editor” section, and submit your "Accept" recommendation.

Reviewer #2: All comments have been addressed

2. Is the manuscript technically sound, and do the data support the conclusions?

Reviewer #2: Yes

3. Has the statistical analysis been performed appropriately and rigorously? 

Reviewer #2: Yes

4. Have the authors made all data underlying the findings in their manuscript fully available?

Reviewer #2: Yes

5. Is the manuscript presented in an intelligible fashion and written in standard English?

Reviewer #2: Yes

6. Review Comments to the Author

Reviewer #2: The author has diligently addressed the comments and suggestions I provided. After reviewing the revisions, I am pleased to say that the work is now accepted in its current form.

7. PLOS authors have the option to publish the peer review history of their article (what does this mean?). If published, this will include your full peer review and any attached files.

Reviewer #2: No

---

## [Editor Report · Acceptance letter]

13 Nov 2023

PONE-D-23-11062R1 

Privacy and Security Concerns with Passively Collected Location Data for Digital Contact Tracing Among U.S. College Students 

Dear Dr. Stevens:

I'm pleased to inform you that your manuscript has been deemed suitable for publication in PLOS ONE. Congratulations! Your manuscript is now with our production department. 

Kind regards, 

on behalf of

Dr. Shadab Alam 

Academic Editor

PLOS ONE